

# FrameAxis: characterizing microframe bias and intensity with word embedding

Haewoon Kwak[1], Jisun An[1], Elise Jing[2] and Yong-Yeol Ahn[2,3,4]

[1] School of Computing and Information Systems, Singapore Management University, Singapore, Singapore
[2] Center for Complex Networks and Systems Research, Luddy School of Informatics, Computing, and Engineering, Indiana University at Bloomington, Bloomington, IN, United States of America
[3] Indiana University Network Science Institute, Indiana University at Bloomington, Bloomington, IN, United States of America
[4] Connection Science, Massachusetts Institute of Technology, Cambridge, MA, United States of America

## ABSTRACT

Framing is a process of emphasizing a certain aspect of an issue over the others, nudging readers or listeners towards different positions on the issue even without making a biased argument. Here, we propose FrameAxis, a method for characterizing documents by identifying the most relevant semantic axes ("microframes") that are overrepresented in the text using word embedding. Our unsupervised approach can be readily applied to large datasets because it does not require manual annotations. It can also provide nuanced insights by considering a rich set of semantic axes. FrameAxis is designed to quantitatively tease out two important dimensions of how microframes are used in the text. *Microframe bias* captures how biased the text is on a certain microframe, and *microframe intensity* shows how prominently a certain microframe is used. Together, they offer a detailed characterization of the text. We demonstrate that microframes with the highest bias and intensity align well with sentiment, topic, and partisan spectrum by applying FrameAxis to multiple datasets from restaurant reviews to political news. The existing domain knowledge can be incorporated into FrameAxis by using custom microframes and by using FrameAxis as an iterative exploratory analysis instrument. Additionally, we propose methods for explaining the results of FrameAxis at the level of individual words and documents. Our method may accelerate scalable and sophisticated computational analyses of framing across disciplines.

Corresponding author
Haewoon Kwak, hkwak@smu.edu.sg

## INTRODUCTION

Framing is a process of highlighting a certain aspect of an issue to make it salient (*Entman, 1993*; *Chong & Druckman, 2007*). By focusing on a particular aspect over another, even without making any biased argument, a biased understanding of the listeners can be induced (*Kahneman & Tversky, 1979*; *Entman, 1993*; *Goffman, 1974*). For example, when reporting on the issue of poverty, a news media may put an emphasis on how successful individuals succeeded through hard work. By contrast, another media may emphasize the failure of national policies. It is known that these distinct framings can induce

contrasting understanding and attitudes about poverty (*Iyengar, 1994*). While readers who are exposed to the former framing became more likely to blame individual failings, those who are exposed to the latter framing tended to criticize the government or other systematic factors rather than individuals. Framing has been actively studied, particularly in political discourse and news media, because framing is considered to be a potent tool for political persuasion (*Scheufele & Tewksbury, 2007*). It has been argued that the frames used by politicians and media shape the public understanding of issue salience (*Chong & Druckman, 2007*; *Kinder, 1998*; *Lakoff, 2014*; *Zaller, 1992*), and politicians strive to make their framing more prominent among the public (*Druckman & Nelson, 2003*).

Framing is not confined to politics. It has been considered crucial in marketing (*Maheswaran & Meyers-Levy, 1990*; *Homer & Yoon, 1992*; *Grewal, Gotlieb & Marmorstein, 1994*), public health campaigns (*Rothman & Salovey, 1997*; *Gallagher & Updegraff, 2011*), and other domains (*Pelletier & Sharp, 2008*; *Huang et al., 2015*). Yet, the operationalization of framing is inherently vague (*Scheufele, 1999*; *Sniderman & Theriault, 2004*) and remains a challenging open question. Since framing research heavily relies on manual efforts from choosing an issue to isolating specific attitudes, identifying a set of frames for an issue, and analyzing the content based on a developed codebook (*Chong & Druckman, 2007*), it is not only difficult to avoid an issue of subjectivity but also challenging to conduct a large-scale, systematic study that leverages huge online data.

Several computational approaches have been proposed to address these issues. They aim to characterize political discourse, for instance, by recognizing political ideology (*Sim et al., 2013*; *Bamman & Smith, 2015*) and sentiment (*Pla & Hurtado, 2014*), or by leveraging established ideas such as the moral foundation theory (*Johnson & Goldwasser, 2018*; *Fulgoni et al., 2016*), general media frame (*Card et al., 2015*; *Kwak, An & Ahn, 2020*), and frame-related language (*Baumer et al., 2015*). Yet, most studies still rely on small sets of predefined ideas and annotated datasets.

To overcome these limitations, we propose FrameAxis, an unsupervised method for characterizing texts with respect to a variety of *microframes*. Each microframe is operationalized by an antonym pair, such as *legal–illegal*, *clean–dirty*, or *fair–unfair*. The value of antonym pairs in characterizing the text has been repeatedly demonstrated (*Haidt & Graham, 2007*; *Johnson & Goldwasser, 2018*; *Fulgoni et al., 2016*; *An, Kwak & Ahn, 2018*; *Kozlowski, Taddy & Evans, 2019*; *Mathew et al., 2020*). For example, MFT identifies the five basic moral 'axes' using antonyms, such as 'Care/Harm' and 'Fairness/Cheating', 'Loyalty/Betrayal', 'Authority/Subversion', and 'Purity/Degradation', as the critical elements for individual judgment (*Haidt & Graham, 2007*). MFT has been applied to discover politicians' stances on issues (*Johnson & Goldwasser, 2018*) and political leaning in partisan news (*Fulgoni et al., 2016*), demonstrating flexibility and interpretability of antonymous semantic axes in characterizing the text. On the other hand, SemAxis (*An, Kwak & Ahn, 2018*) and following studies (*Kozlowski, Taddy & Evans, 2019*; *Mathew et al., 2020*) leverage the word embeddings to characterize the semantics of a word in different communities or domains (e.g., different meaning of 'soft' in the context of sports vs. toy) by computing the similarities between the word and a set of predefined antonymous axes

("semantic axes"). As in SemAxis, FrameAxis leverages the power of word embedding, which allows us to capture similarities between a word and a semantic axis.

For each microframe defined by an antonym pair, FrameAxis is designed to quantitatively tease out two important dimensions of how it is used in the text. *Microframe bias* captures how biased the text is on a certain microframe, and *microframe intensity* shows how prominently a certain microframe is used. Both dimensions together offer a nuanced characterization of the text. For example, let us explain the framing bias and intensity of the text about an immigration issue on the *illegal–legal* microframe. Then, the framing bias measures how much the text focuses on an 'illegal' perspective of the immigration issue rather than a 'legal' perspective (and vice versa); the framing intensity captures how much the text focuses on an illegal *or* legal perspective of the immigration issue rather than other perspectives, such as segregation (i.e., *segregated–desegregated* microframe).

While FrameAxis works in an unsupervised manner, FrameAxis can also benefit from manually curated microframes. When domain experts are already aware of important candidate frames of the text, they can be directly formulated as microframes. For the case when FrameAxis works in an unsupervised manner—which would be much more common, we propose methods to identify the most relevant semantic axes based on the values of microframe bias and intensity. Moreover, we also suggest document and word-level analysis methods that can explain *how* and *why* the resulting microframe bias and intensity are found with different granularity.

We emphasize that FrameAxis cannot replace conventional framing research methods, which involves sophisticated close reading of the text. Also, we do not expect that the microframes can be directly mapped to the frames identified by domain experts. FrameAxis can thus be considered as a computational aid that can facilitate systematic exploration of texts and subsequent in-depth analysis.

## METHODS

FrameAxis involves four steps: (i) compiling a set of microframes, (ii) computing word contributions to each microframe, (iii) calculating microframe bias and intensity by aggregating the word contributions, and finally (iv) identifying significant microframes by comparing with a null model. We then present how to compute the relevance of microframes to a given corpus.

### Building a set of predefined microframes

FrameAxis defines a microframe as a "semantic axis" (*An, Kwak & Ahn, 2018*) in a word vector space—a vector from one word to its antonym. Given a pair of antonyms (pole words), $w^+$ (e.g., 'happy') and $w^-$ (e.g., 'sad'), the semantic axis vector is $v_f = v_{w^+} - v_{w^-}$, where $f$ is a microframe or a semantic axis (e.g., *happy–sad*), and $v_{w^+}$ and $v_{w^-}$ are the corresponding word vectors. To capture nuanced framing, it is crucial to cover a variety of antonym pairs. We extract 1,828 adjective antonym pairs from WordNet (*Miller, 1995*) and remove 207 that are not present in the GloVe embeddings (840B tokens, 2.2M vocab, 300d vectors) (*Pennington, Socher & Manning, 2014*). As a result, we use 1,621 antonym

pairs as the predefined microframes. As we explained earlier, when potential microframes of the text are known, using only those microframes is also possible.

## Computation of microframe bias and intensity

A microframe $f$ (or semantic axis in (*An, Kwak & Ahn, 2018*)) is defined by a pair of antonyms $w^+$ and $w^-$. Microframe bias and intensity computation are based on the contribution of each word to a microframe. Formally, we define the contribution of a word $w$ to a microframe $f$ as the similarity between the word vector $v_w$ and the microframe vector $v_f$ $(= v_{w+} - v_{w-})$. While any similarity measure between two vectors can be used here, for simplicity, we use cosine similarity:

$$c_f^w = \frac{v_w \cdot v_f}{\| v_w \| \| v_f \|} \tag{1}$$

We then define microframe bias of a given corpus $t$ on a microframe $f$ as the weighted average of the word's contribution $c_f^w$ to the microframe $f$ for all the words in $t$. This aggregation-based approach shares conceptual roots with the traditional expectancy value model (*Nelson, Oxley & Clawson, 1997b*), which explains an individual's attitude to an object or an issue. In the model, the individual's attitude is calculated by the weighted sum of the evaluations on attribute $a_i$, whose weight is the salience of the attribute $a_i$ of the object. In FrameAxis, a corpus is represented as a bag of words, and each word is considered an attribute of the corpus. Then, a word's contribution to a microframe can be considered as the evaluation on attribute, and the frequency of the word can be considered as the salience of an attribute. Accordingly, the weighted average of the word's contribution to the microframe $f$ for all the words in $t$ can be mapped onto the individual's attitude toward an object—that is, microframe bias. An analogous framework using a weighted average of each word's score is also proposed for computing the overall valence score of a document (*Dodds & Danforth, 2010*). Formally, we calculate the microframe bias, $B_f^t$, of a text corpus $t$ on a microframe $f$ as follows:

$$B_f^t = \frac{\sum_{w \in t} (n_w c_f^w)}{\sum_{w \in t} n_w} \tag{2}$$

where $n_w$ is the number of occurrences of word $w$ in $t$.

Microframe intensity captures how strongly a given microframe is used in the document. Namely, given corpus $t$ on a microframe $f$ we measure the second moment of the word contributions $c_f^w$ on the microframe $f$ for all the words in $t$. For instance, if a given document is emotionally charged with many words that strongly express either happiness or sadness, we can say that the *happy–sad* microframe is heavily used in the document regardless of the microframe bias regarding the *happy–sad* axis.

Formally, microframe intensity, $I_f^t$, of a text corpus $t$ on a microframe $f$ is calculated as follows:

$$I_f^t = \frac{\sum_{w \in t} n_w (c_f^w - B_f^T)^2}{\sum_{w \in t} n_w} \tag{3}$$

where $B_f^T$ is the baseline microframe bias of the entire text corpus $T$ on a microframe $f$ for computing the second moment. As the squared term is included in the equation, the words

that are far from the baseline microframe bias—and close to either of the poles—contribute strongly to the microframe intensity.

We present an illustration of microframe intensity and bias in Fig. 1A, where arrows represent the vectors of words appeared in a corpus, and blue and orange circles represent two pole word vectors, which define the $w^+$–$w^-$ microframe. If words that are semantically closer to one pole are frequently used in a corpus, the corpus has the high microframe bias toward that pole and the high microframe intensity on the $w^+$–$w^-$ microframe (top right). By contrast, if words that are semantically closer to both poles are frequently used, the overall microframe bias becomes low by averaging out the biases toward both poles, but the microframe intensity stays high because the $w^+$–$w^-$ microframe is actively used (bottom right).

### Handling non-informative topic words

It is known that pretrained word embeddings have multiple biases (*Bolukbasi et al., 2016*). Although some de-biasing techniques are proposed (*Zhao et al., 2018*), those biases are not completely eliminated (*Gonen & Goldberg, 2019*). For example, the word 'food' within a GloVe pretrained embedding space is much closer to 'savory' (cosine similiarity: 0.4321) than 'unsavory' (cosine similarity: 0.1561). As those biases could influence the framing bias and intensity due to its high frequencies in the text of reviews on food, we remove it from the analysis of the reviews on food.

FrameAxis computes the word-level framing bias (intensity) shift to help this process, which we will explain in 'Explainability' Section. Through the word-level shift, FrameAxis users can easily check whether some words that should be neutral on a certain semantic axis are located as neutral within a given embedding space.

While this requires manual efforts, one shortcut is to check the topic word first. For example, when FrameAxis is applied to reviews on movies, the word 'movie' could be considered first because 'movie' should be neutral and non-informative. Also, as reviews on movies are likely to contain the word 'movie' multiple times, even smaller contribution of 'movie' to a given microframe could be amplified by its high frequency of occurrences, $n_w$ in Eqs. (2) and (3). After the manual confirmation, those words are replaced with <UNK> tokens and are not considered in the computation of framing bias and intensity.

In this work, we also removed topics words as follows: in the restaurant review dataset, the word indicating aspect (i.e., *ambience, food, price,* and *service*) is replaced with <UNK> tokens. In the AllSides' political news dataset, we consider the issue defined by AllSides as topic words, such as *abortion, immigration, elections, education, polarization,* and so on.

### Identifying statistically significant microframes

The microframe bias and intensity of a target corpus can be interpreted with respect to the background distribution for statistical significance. We compute microframe bias and intensity on the microframe $f$ from a bootstrapped sample $s$ from the entire corpus $T$, denote by $B_f^{NULL_s}$ and $I_f^{NULL_s}$, respectively. We set the size of the sample $s$ to be equal to that of the target corpus $t$.

Then, the differences between $B_f^{NULL_s}$ and $B_f^t$ and that between $I_f^{NULL_s}$ and $I_f^t$ shows how likely the microframe bias and intensity in the target corpus can be obtained by chance. The

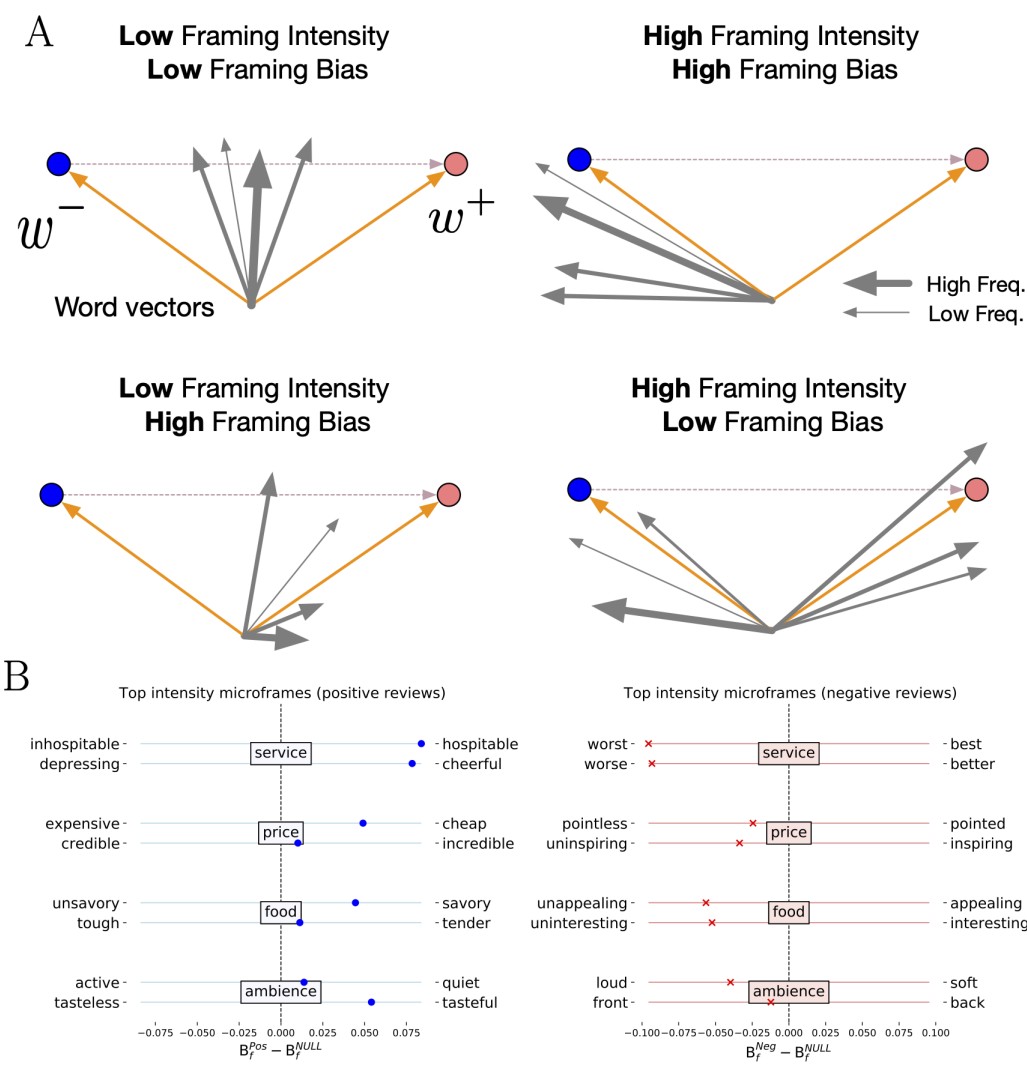

**Figure 1** **Illustrations of framing intensity and framing bias and their values in restaurant reviews.**
(A) Illustrations of microframe intensity and framing bias. Blue and orange circles represent two pole
word vectors, which define the $w^+$—$w^-$ microframe, and gray arrows represent the vector of words ap-
peared in a given corpus. The width of the arrows indicates the weight (i.e., frequency of appearances) of
the corresponding words. The figure shows when microframe intensity and bias can be high or low. (B)
Microframe bias with respect to the top two microframes with the highest microframe intensity for each
aspect in restaurant reviews. The high-intensity microframes are indeed relevant to the corresponding as-
pect, and microframe biases on these microframes are also consistent with the sentiment labels.

statistical significance of the observation is calculated by doing two-tailed tests on the $N$
bootstrap samples. By setting a threshold $p$-value, we identify the significant microframes.
In this work, we use $N = 1,000$ and $p = 0.05$.

We also can compute the effect size ($|\eta|$, which is the difference between the observed
value and the sample mean) for microframe $f$:

$$\eta_f^{\mathrm{B}} = \mathrm{B}_f^t - \mathrm{B}_f^{NULL} = \mathrm{B}_f^t - \frac{\sum_i^N \mathrm{B}_f^{NULL_{s_i}}}{N} \tag{4}$$

$$\eta_f^{\mathrm{I}} = \mathrm{I}_f^t - \mathrm{I}_f^{NULL} = \mathrm{I}_f^t - \frac{\sum_i^N \mathrm{I}_f^{NULL_{s_i}}}{N} \tag{5}$$

We can identify the top $M$ significant microframes in terms of the microframe bias (intensity) by the $M$ microframes with the largest $|\eta^{\mathrm{B}}|$ ($|\eta^{\mathrm{I}}|$).

## Microframe bias and intensity shift per word

We define the word-level microframe bias and intensity shift in a given corpus $t$ as follows:

$$S_w^t(\mathrm{B}_f) = \frac{n_w c_f^w}{\sum_{w \in t} n_w} \tag{6}$$

$$S_w^t(\mathrm{I}_f) = \frac{n_w (c_a^w - \mathrm{B}_f^T)^2}{\sum_{w \in t} n_w} \tag{7}$$

which shows how a given word ($w$) brings a shift to microframe bias and intensity by considering both the word's contribution to the microframe ($c_a^w$) and its appearances in the target corpus $t$ ($n_w$). In this work, both shifts are compared to those from the background corpus.

## Contextual relevance of microframes to a given corpus

Not all predefined microframes are necessarily meaningful for a given corpus. While we provide a method to compute the statistical significance of each microframe for a given corpus, filtering out irrelevant microframes in advance can reduce computation cost. We propose two methods to compute relevance of microframes to a given corpus: embedding-based and language model-based approaches.

First, an embedding-based approach calculates relevance of microframes as cosine similarity between microframes and a primary topic of the corpus within a word vector space. A topic can be defined as a set of words related to a primary topic of the corpus. We use $\tau = \{w_{t1}, w_{t2}, w_{t3}, \ldots, w_{tn}\}$ to represent a set of topic words. The cosine similarity between a microframe $f$ defined by two pole words $w^+$ and $w^-$ and a set of topic words $\tau$ can be represented as the average cosine similarity between pole word vectors ($v_{w^+}$ and $v_{w^-}$) and a topic word vector ($v_{w_{ti}}$):

$$
\begin{aligned}
r_f^t &= \frac{1}{|\tau|} \sum_{w_{ti} \in \tau} \frac{(\text{relevance of } w^+ \text{ to } w_{ti}) + (\text{relevance of } w^- \text{ to } w_{ti})}{2} \\
&= \frac{1}{|\tau|} \sum_{w_{ti} \in \tau} \frac{\frac{v_{w_{ti}} \cdot v_{w^+}}{\|v_{w_{ti}}\| \|v_{w^+}\|} + \frac{v_{w_{ti}} \cdot v_{w^-}}{\|v_{w_{ti}}\| \|v_{w^-}\|}}{2}
\end{aligned}
\tag{8}
$$

Second, a language model-based approach calculates relevance of microframes as perplexity of a template-filled sentence. For example, consider two templates as follows:

- T1(topic word, pole word): {topic word} is {pole word}.

- T2(topic word, pole word): {topic word} are {pole word}.

If a topic word is 'healthcare' and a microframe is *essential–inessential*, four sentences, which are 2 for each pole word, can be generated. Following a previous method (*Wang & Cho, 2019*), we use a pre-trained OpenAI GPT model to compute the perplexity score.

We take a lower perplexity score for each pole word because a lower perplexity score should be from the sentence with a correct subject-verb pair (i.e., singular-singular or plural-plural). In this stage, for instance, we take 'healthcare is essential' and 'healthcare is inessential'. Then, we sum two perplexity scores from one pole and the other pole words and call it frame relevance of the corresponding microframe to the topic.

According to the corpus and topic, a more complex template, such as "A (an) {topic word} issue has a {pole word} perspective." might work better. More appropriate template sentences can be built with good understanding of the corpus and topic.

### Human evaluation

We perform human evaluations through Amazon Mechanical Turk (MTurk). For microframe bias, we prepare the top 10 significant microframes ranked by the effect size (i.e., answer set) and randomly selected 10 microframes with an arbitrary microframe bias (i.e., random set) for each pair of aspect and sentiment (e.g., *positive* reviews about *ambience*). As it is hard to catch subtle differences of the magnitude of microframe biases and intensity through crowdsourcing, we highlight microframe bias on each microframe with bold-faced instead of its numeric value.

As a unit of question-and-answer tasks in MTurk (Human Intelligence Task [HIT]), we ask "Which set of antonym pairs do better characterize a *positive* restaurant review on *ambience*? (A word on the right side of each pair (in bold) is associated with a *positive* restaurant review on *ambience*.)" The italic text is changed according to every aspect and sentiment. We note that, for every HIT, the order of microframes in both sets is shuffled. The location (i.e., top or bottom) of the answer set is also randomly chosen to avoid unexpected biases of respondents.

For microframe intensity, we prepare the top 10 significant microframes (i.e., answer set) and randomly selected 10 microframes (i.e., random set) for each pair of aspect and sentiment. The top 10 microframes are chosen by the effect size, computed by eq4,eq5, among the significant microframes. We then ask "Which set of antonym pairs do better characterize a *positive* restaurant review on *service*?" The rest of the procedure is the same as the framing bias experiment.

For the quality control of crowd-sourced answers, we recruit workers who (1) live in the U.S., (2) have more than 1,000 approved HITs, and (3) achieve 95% of approval rates. Also, we allow a worker to answer up to 10 HITs. We recruit 15 workers for each (aspect, sentiment) pair. We pay 0.02 USD for each HIT.

## RESULTS

### Microframe in restaurant reviews

To validate the concept of microframe bias and intensity, we examine the SemEval 2014 task 4 dataset (*Pontiki et al., 2014*), which is a restaurant review dataset where reviews

are grouped by aspects (food, ambience, service, and price) and sentiment (positive and negative). This dataset provides an ideal playground because (i) restaurant reviews tend to have a clear bias—whether the experience was good or bad—which can be used as a benchmark for framing bias, and ii) the aspect labels also help us perform the fine-grained analysis and compare microframes used for different aspects of restaurant reviews.

We compute microframe bias and intensity for 1,621 predefined microframes, which are compiled from WordNet (*Miller, 1995*) (See *Methods* for detail), for every review divided by aspects and sentiments. The top two microframes with the highest microframe intensity are shown in Fig. 1. For each highest-intensity microframe, we display the microframe bias that is computed through a comparison between the positive (negative) reviews and the null model—bootstrapped samples from the whole corpus (See *Methods*). The highest-intensity microframes are indeed relevant to the corresponding aspect: *hospitable–inhospitable* and *best–worst* for service, *cheap–expensive* and *pointless–pointed* for price, *savory–unsavory* and *appealing–unappealing* for food, and *active–quiet* and *loud–soft* for ambience. At the same time, it is clear that positive and negative reviews tend to focus on distinct perspectives of the experience. Furthermore, observed microframe biases are consistent with the sentiment labels; microframe biases in positive reviews are leaning toward the positive side of the microframes, and those in negative reviews toward the negative side.

In other words, FrameAxis is able to automatically discover that positive reviews tend to characterize service as *hospitable*, price as *cheap*, food as *savory*, and ambience is *tasteful*, and negative reviews describe service as *worst*, price as *pointless*, food as *unappealing*, and ambience as *loud* in an unsupervised manner. Then, how and why do these microframes get those bias and intensity? In the next section, we propose two tools to provide explainability with different granularity behind microframe bias and intensity.

## Explainability

To understand computed microframe bias and intensity better, we propose two methods: (i) word-level microframe bias (intensity) shift, and (ii) document-level microframe bias (intensity) spectrum.

The word-level impact analysis has been widely used for explaining results of the text analysis (*Dodds & Danforth, 2010*; *Ribeiro, Singh & Guestrin, 2016*; *Lundberg & Lee, 2017*; *Gallagher et al., 2020*). Similarly, we can compute the word-level microframe shift that captures how each word in a target corpus $t$ influences the resulting microframe bias (intensity) by aggregating contributions of the word $w$ for microframe bias (intensity) on microframe $f$ (See *Methods*). It is computed by comparison with its contribution to a background corpus. For instance, even though $w$ is a word that conveys positive connotations, its contribution to a target corpus can become negative if its appearance in $t$ is lower than that in the background corpus.

Figure 2A shows the top 10 words with the highest microframe bias shift for the two high-intensity microframes from the 'food' aspect. On the left, the green bars show how each word in the positive reviews shifts the microframe bias toward either savory or unsavory on the *savory–unsavory* microframe, and the gray bars show how the same word in the background corpus (non-positive reviews) shifts the microframe bias. The difference

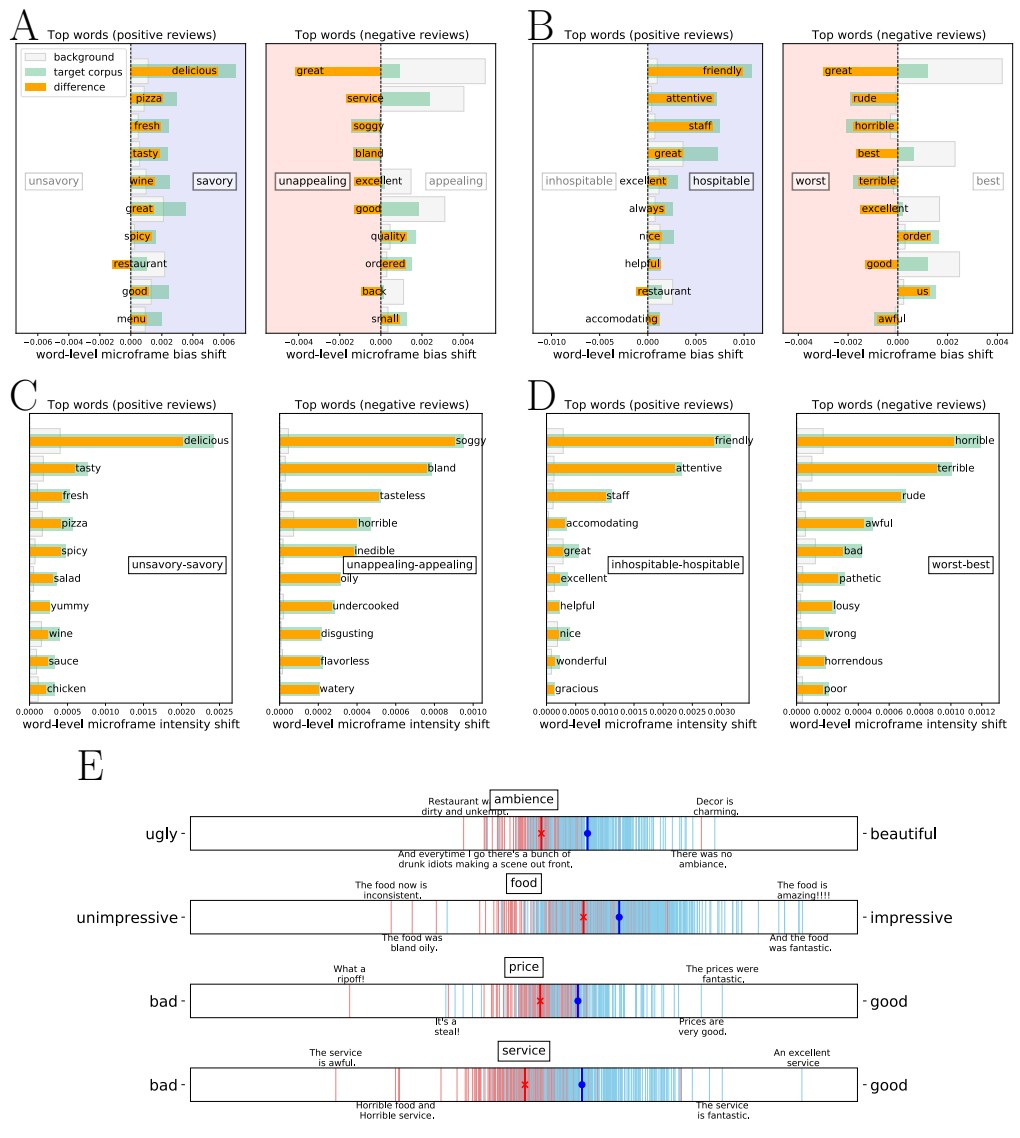

**Figure 2** **Word-level microframe shift diagram and microframe spectrum for understanding framing intensity and bias.** (A–D) Word-level contribution to selected microframes in restaurant reviews shows which words contribute to the resulting microframe bias and intensity the most. (E) Document-level microframe bias spectra of positive (blue) and negative (red) reviews about different aspects. Two reviews about service, 'An excellent service' and 'The service is fantastic' have the microframe bias closer to the 'good' on the bad—good microframe, and other two, 'The service is awful' and 'Horrible food and Horrible service' have the microframe bias closer to the 'bad'. These spectra clearly show the microframe bias that each document has on a given microframe.

between the two shifts, which is represented as the orange bars, shows the effect of each word for microframe bias on the *savory–unsavory* microframe in positive reviews. The same word's total contribution differs due to the frequency because its contribution on the axis $c_f^w$ is the same. For instance, the word 'delicious' appears 80 times in positive reviews, and the normalized term frequency is 0.0123. By contrast, 'delicious' appears

only three times in non-positive reviews, and the normalized term frequency is 0.0010. In short, the normalized frequency of the word 'delicious' is an order of magnitude higher in positive reviews than non-positive reviews, and thus the difference strongly shifts the microframe bias toward 'savory' on the *savory–unsavory* microframe. A series of the words describing positive perspectives of food, such as *delicious, fresh, tasty, great, good, yummy,* and *excellent,* appears as the top words with the highest microframe bias shifts toward savory on the *savory–unsavory* microframe.

Similarly, on the right in Fig. 2A, the green and the gray bars show how each word in the negative reviews and background corpus (non-negative reviews) shifts the microframe bias on the *appealing–unappealing* microframe, respectively, and the orange bar shows the difference between the two shifts. The word 'great' in the negative reviews shifts microframe bias toward appealing less than that in the background corpus; the word 'great' less frequently appears in negative reviews (0.0029) than in background corpus (0.0193). Consequently, the resulting microframe bias attributed from the word 'great' in the negative reviews is toward 'unappealing' on the *appealing–unappealing* microframe. In addition, words describing negative perspectives of food, such as *soggy, bland, tasteless, horrible,* and *inedible*, show the orange bars heading to unappealing rather than appealing side on the *appealing –unappealing* microframe. Note that the pole words for these microframes do not appear in the top word lists. These microframes are found because they best capture—according to word embedding—these words *collectively*.

Figure 2C shows the top 10 words with the highest framing intensity shift for the two high-intensity microframes from the 'food' aspect. On the right, compared to Fig. 2A, more words reflecting the nature of the *unappealing–appealing* microframe, such as *tasteless, horrible, inedible, oily, undercooked, disgusting, flavorless,* and *watery,* are shown as top words in terms of the microframe intensity shift. As we mentioned earlier, we confirm that the words that are far from the baseline framing bias—and close to either of the poles—contribute strongly to the microframe intensity.

Figure 2B shows another example of word-level framing bias shift in the reviews about service. On the left, top words that shift microframe bias toward 'hospitable', such as *friendly, attentive, great, excellent, nice, helpful, accommodating, wonderful,* and *prompt*, are captured from the positive reviews. On the right, top words that shift microframe bias toward 'worst', such as *rude, horrible, terrible, awful, bad, wrong,* and *pathetic*, are found. Similar to the top words in negative reviews about food, some words that shift framing bias toward 'best' less frequently appear in the negative reviews than the background corpus, making their impact on microframe bias be farther from 'best' on the *best –worst* microframe.

As the word-level microframe shift diagram captures, what FrameAxis detects is closely linked to the abundance of certain words. Does it mean that our results merely reproduce what simpler methods for detecting overrepresented words perform? To answer this question, we compare the log odds ratio with informative Dirichlet prior (*Monroe, Colaresi & Quinn, 2017*). With the log odds ratio, *service, friendly, staff, attentive, prompt, fast, helpful, owner*, and *always* are found to be overrepresented words. This list of overrepresented words is always the *same* when comparing given corpora because it only considers their frequencies

of appearances in the corpora. By contrast, FrameAxis identifies the most relevant words for each microframe by considering their appearances and their contributions to the microframe, providing richer interpretability. Even though a word appears many times in a given corpus, it does not shift the microframe bias or intensity if the word is irrelevant to the microframe.

Figure 2D shows the top 10 words with the highest microframe intensity shift for the two high-intensity microframes from the 'service' aspect. On the left, compared to Fig. 2B, more words describing the *inhospitable–hospitable* microframe, such as *wonderful* and *gracious*, are included. On the right, compared to Fig. 2B, more words reflecting the nature of the *worst–best* microframe, such as *bad, pathetic, lousy, wrong, horrendous,* and *poor* are shown as top words in terms of the framing intensity shift.

The second way to provide explainability is computing a document-level framing bias (intensity) and visualizing them as a form of a microframe bias (intensity) spectrum. Figure 2E shows microframe bias spectra of positive and negative reviews. Each blue and red line corresponds to an individual positive and negative review, respectively. Here we choose microframes that show large differences of microframe bias between the positive and negative reviews as well as high intensities by using the following procedure. We rank microframes based on average microframe intensity across the reviews and the absolute differences of microframe bias between the positive and negative reviews, sum both ranks, and pick the microframes with the lowest rank-sum for each aspect. In contrast to the corpus-level microframe analysis or word-level microframe shift, this document-level microframe analysis provides a mesoscale view showing where each document locates on the microframe bias spectrum.

## Microframe bias and intensity separation

As we show that FrameAxis reasonably captures relevant microframe bias and intensity from positive reviews and negative reviews, now we focus on how the most important dimension of positive and negative reviews—positive sentiment and negative sentiment—is captured by FrameAxis. Consider that there is a microframe that can be mapped into a sentiment of reviews. Then, if FrameAxis works correctly, microframe biases on the corresponding microframe captured from positive reviews and negative reviews should be significantly different.

Formally, we define a *microframe bias separation* as the difference between microframe bias of positive reviews and that of negative reviews on microframe $f$, which is denoted by $\Delta_{\text{B}_f}^{pos-neg} = $ (Microframe bias on microframe $f$ of positive reviews) $-$ (Microframe bias on microframe $f$ of negative reviews) $= \text{B}_f^{pos} - \text{B}_f^{neg}$. Similarly, a *microframe intensity separation* can be defined as following: $\Delta_{\text{I}_f}^{pos-neg} = $ (Microframe intensity on microframe $f$ of positive reviews) - (Microframe intensity on microframe $f$ of negative reviews) $= \text{I}_f^{pos} - \text{I}_f^{neg}$.

Figure 3 shows the cumulative density function (CDF) of the magnitude of microframe bias separations, $|\Delta_{\text{B}_f}^{pos-neg}|$, for 1,621 different microframes for each aspect. Given that the *bad–good* axis is a good proxy for sentiment (*An, Kwak & Ahn, 2018*), the *bad–good* microframe would have a large bias separation *if* the microframe bias on that microframe

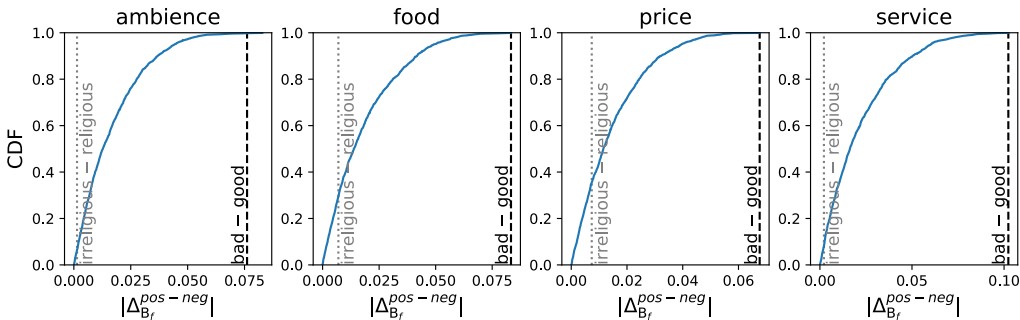

**Figure 3** **CDF of the magnitude of a microframe bias separation, which is the difference of average microframe biases between positive and negative reviews.** As we can expect from the nature of the dataset (positive and negative reviews), the *good—bad* microframe, which maps into a sentiment axis, exhibits a large microframe bias separation. By contrast, *irreligious—religious* frame, which is rather irrelevant in restaurant reviews, shows a small microframe bias separation as expected.

captures the sentiment correctly. Indeed, the *bad–good* microframe shows a large separation –larger than 99.91 percentile across all aspects (1.5th rank on average). For comparison, the *irreligious –religious* microframe does not separate positive and negative restaurant reviews well (19.88 percentile, 1,298.8th rank on average). The large microframe bias separation between the microframe bias of positive reviews and that of negative reviews supports that the *bad –good* microframe—and thus FrameAxis—captures the most salient dimension of the text.

Using the two separation measures, we can compare two corpora with respect to both microframe intensity and bias. We find that the absolute values of both separations, $|\Delta_{I_f}^{pos-neg}|$ and $|\Delta_{B_f}^{pos-neg}|$, are positively correlated across the four aspects (Spearman's correlation $\rho = 0.379$ (ambience), 0.471 (food), 0.228 (price), and 0.304 (service)), indicating that when a certain microframe is more heavily used, it also tends to be more strongly biased.

To illustrate a detailed picture, we show microframe intensity and bias separation of each microframe in Fig. 4. Microframes above the gray horizontal line have higher microframe intensity in positive reviews than negative reviews. We indicate the microframe bias with bold face. **Word**[+] indicates that positive reviews are biased toward the pole, and **word**[−] means the opposite (negative reviews). For instance, at the top, the label 'sour-**sweet**[+]' indicates that 'sweetness' of the ambience is highlighted in positive reviews, and the label '**loud**[−]-soft' indicates that 'loudness' of the ambience frequently appears in negative reviews. For clarity, the labels for microframes are written for top 3 and bottom 3 microframes of $\Delta_{I_f}^{pos-neg}$ and $\Delta_{B_f}^{pos-neg}$ each.

This characterization provides a comprehensive view of how microframes are employed in positive and negative reviews for highlighting different perspectives. For instance, when people write reviews about the price of a restaurant, *incredible, nice, good, cheap, incomparable, best,* and *pleasant* perspectives are highlighted in positive reviews, but *judgmental, unoriginal, pointless,* and *unnecessary* are highlighted in negative reviews. From the document-level framing spectrum analysis, the strongest 'judgmental' and 'unnecessary'

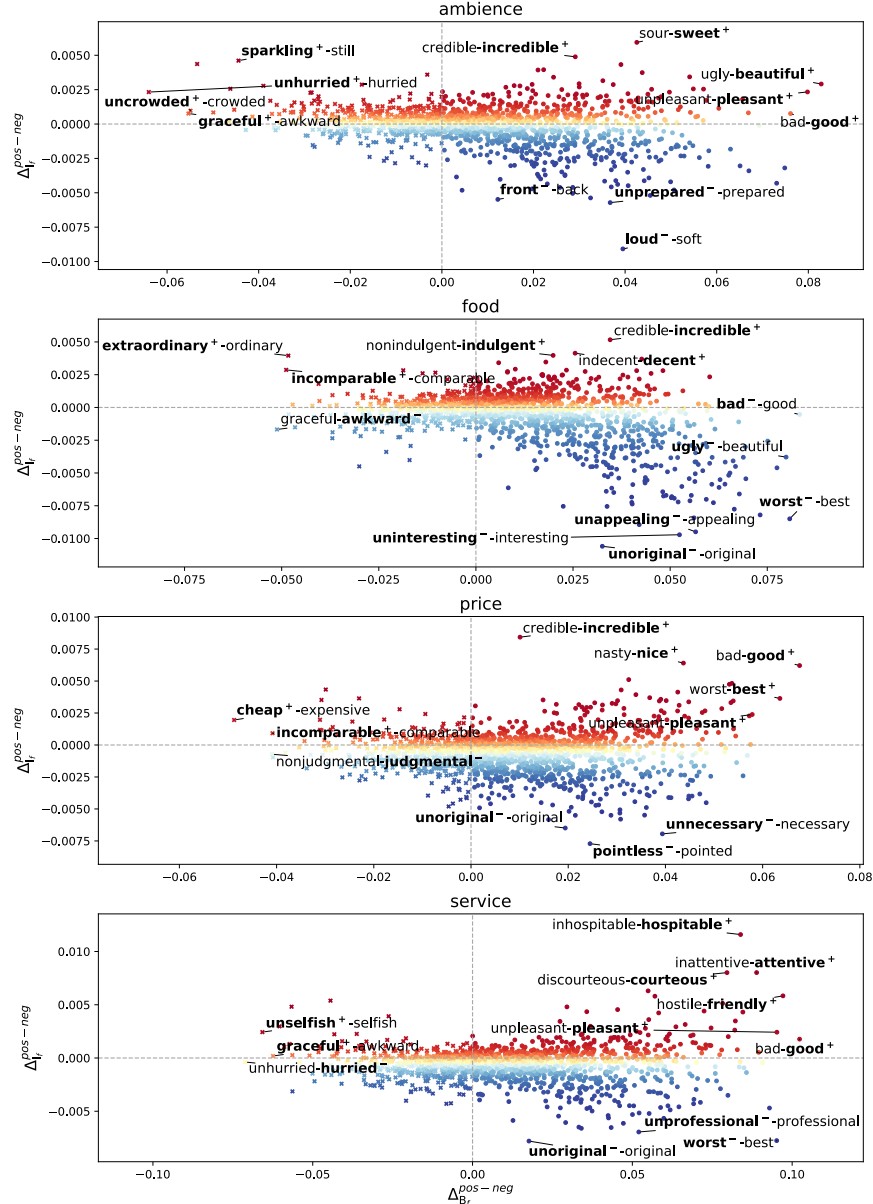

**Figure 4** **Microframe intensity and bias separation of each microframe between positive and negative reviews.** We highlight the microframe bias of positive reviews when $\Delta_{I_f}^{pos-neg} > 0$ ($f$ is more highlighted in positive reviews). Similarly, we highlight the bias of negative reviews when $\Delta_{I_f}^{pos-neg} < 0$ ($f$ is more highlighted in negative reviews). For clarity, we add the subscript to represent the microframe bias of the positive reviews (+) or negative reviews (-). Microframe biases in the positive reviews generally convey positive connotations, and those in the negative reviews convey negative connotations.

microframe biases are found from the reviews about the reasoning behind pricing, such as 'Somewhat pricey but what the heck.'

While some generic microframes, such as *incredible–credible* or *worst–best*, are commonly found across different aspects, aspect-specific microframes, such as *uncrowded–crowded* or

**Table 1 Human evaluation results.** Human evaluation for significant microframes with the top 10 highest framing bias and intensity.

| (Sentiment) Aspects | Accuracy for framing bias | Accuracy for framing intensity |
|---|---|---|
| (+) Service | 1.000 | 0.867 |
| (+) Price | 0.867 | 0.733 |
| (+) Food | 0.933 | 0.800 |
| (+) Ambience | 1.000 | 0.600 |
| (−) Service | 0.867 | 0.867 |
| (−) Price | 0.667 | 0/667 |
| (−) Food | 0.867 | 0.733 |
| (−) Ambience | 0.800 | 0.733 |
| Average | 0.875 | 0.750 |

*inhospitable–hospitable*, are found in the reviews about corresponding aspect. Most of the microframe biases in the positive reviews convey positive connotations, and those in the negative reviews convey negative connotations.

## Human evaluation

We perform human evaluations through Amazon Mechanical Turk (MTurk). Similar with the word intrusion test in evaluating topic modeling (*Chang et al., 2009*), we assess the quality of identified framing bias and intensity by human raters.

For microframe bias, we prepare the top 10 significant microframes with the highest microframe bias (i.e., answer set) and randomly selected 10 microframes with an arbitrary bias (i.e., random set) for each pair of aspect and sentiment (e.g., *positive* reviews about *ambience*). As it is hard to catch subtle differences of the magnitude of microframe biases and intensity through crowdsourcing, we highlight microframe bias on each microframe with bold-faced like Fig. 4 instead of showing its numeric value. We then ask which set of microframe with a highlighted bias do better characterize a given corpus, such as 'positive' reviews on 'ambience'. See *Methods* for detail.

For microframe intensity, we prepare the top 10 significant microframes (i.e., answer set) and randomly selected 10 microframes (i.e., random set) for each pair of aspect and sentiment. We then ask which set of microframe do better characterize a given corpus, such as 'positive' reviews on 'ambience'.

Table 1 shows the fraction of the correct choices of workers (i.e., choosing the answer set). The overall average accuracy is 87.5% and 75.0% for significant microframes with the highest microframe bias and intensity, respectively. For microframe bias, in (+) Service and (+) Ambience, human raters chose the answer sets correctly without errors. By contrast, for microframe intensity, some sets show a relatively lower performance. We manually check them for error analysis and find that workers tended to choose the random set when generic microframes, such as *positive –negative*, appear in a random set due to its ease of interpretation.

### Contextually relevant microframes

As we mentioned in *Method*, in addition to automatically identified microframes that are strongly expressed in a corpus, we can discover microframes that are relevant to a given topic without examining the corpus. We use each aspect—food, price, ambience, and service—as topic words. By using the embedding-based approach, we find *healthy–unhealthy* for food, *cheap–expensive* for price, *noisy–quiet* for ambience, and *private–public* for service as the most relevant microframes. It is also possible to use different words as topic words for identifying relevant microframes. For example, one might be curious about how people think about waiters specifically among the reviews on service. In this case, the most relevant microframes become *impolite–polite* and *attentive–inattentive* by using 'waiter' as a topic word. Then, the computed microframe bias and intensity show how these microframes are used in a given corpus.

## Microframe in political news

As a demonstration of another practical application of FrameAxis, we examine news media. The crucial role of media's framing in public discourse on social issues has been widely recognized (*Nelson, Clawson & Oxley, 1997a*). We show that FrameAxis can be used as an effective tool to characterize news on different issues through microframe bias and intensity. We collect 50,073 news headlines of 572 liberal and conservative media from AllSides (*AllSides, 2012*). These headlines fall in one of predefined issues defined by AllSides, such as *abortion, immigration, elections, education, polarization,* and so on. We examine framing bias and intensity from the headlines for a specific issue, considering all three aforementioned scenarios.

The first scenario is when *domain experts already know which microframes are worth to examine*. For example, news about immigration can be approached through an *illegal –legal* framing (*Wright, Levy & Citrin, 2016*). In this case, FrameAxis can reveal how strong 'illegal vs. legal' media framing is and which position a media outlet has through the microframe bias and intensity on the *illegal–legal* microframe.

Figures 5A–5C show how FrameAxis can capture microframes used in news reporting with different granularity: (A) media-level microframe bias-intensity map, (B) word-level microframe bias shift, and (C) document (news headline)-level microframe bias spectrum. Figure 5A exhibits the average microframe intensity and bias of individual media, which we call a microframe bias-intensity map. To reveal the general tendency of conservative and liberal media's microframe, we also plot their mean on the map. For clarity, we filter out media that have less than 20 news headlines about immigration. Conservative media have higher microframe intensity than liberal media, meaning that they more frequently report the *illegal–legal* microframe of the immigration issue than liberal media do. In addition, conservative media have the microframe bias that is closer to illegal than legal compared to liberal media, meaning that they report more on the illegality of the immigration issue. In summary, the media-level microframe bias-intensity map presents framing patterns of news media on the immigration issue; conservative media do report illegal perspectives more than legal perspectives of the issue and do more frequently report them than liberal media do.

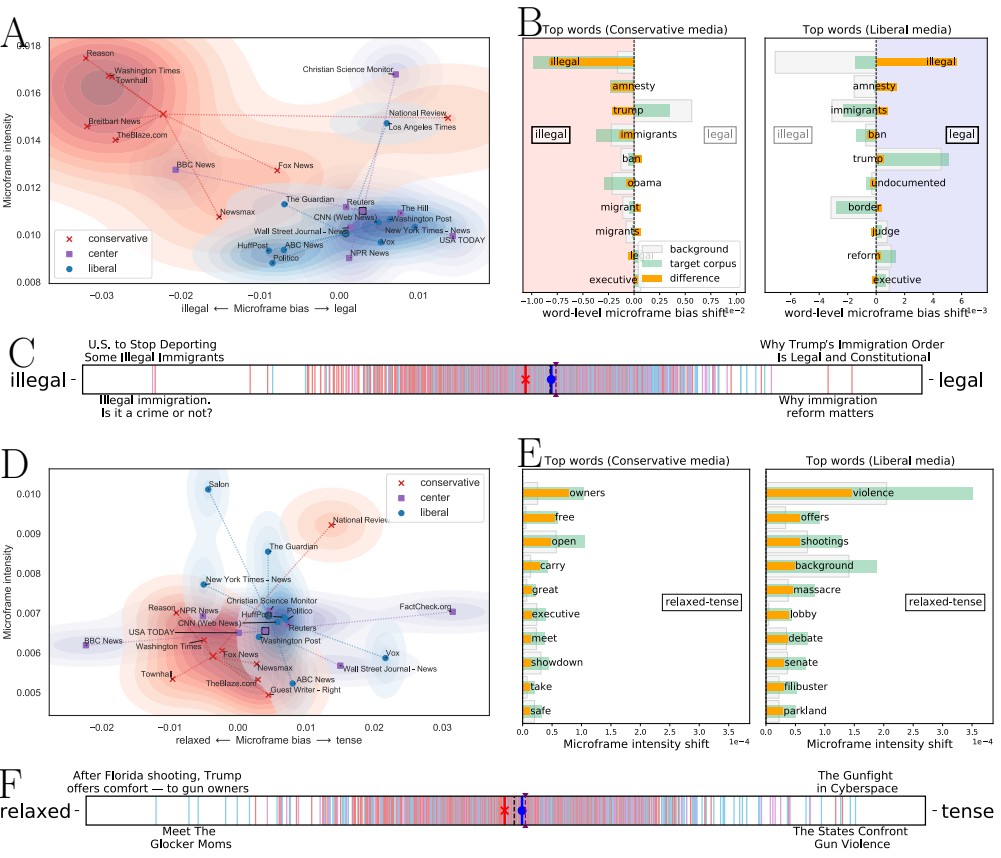

**Figure 5** **Three different views characterizing news headlines for an immigration issue on the *illegal—legal* microframe.** Three different views characterizing media's framing for (A–C) an immigration issue on the *illegal—legal* microframe, and (D–F) gun control and right issue on the *relaxed—tense* microframe. (A) and (D), Media-level microframe bias-intensity map. Each media is characterized by microframe intensity and bias. Conservative media has microframe bias toward 'illegal' while liberal media has microframe bias toward 'legal' in news on an immigration issue. Also, conservative media use the illegal –legal microframe more intensively. (B and E), Word-level microframe bias and intensity shift. Each word contributes the most to the resulting microframe bias. Conservative media use the word 'illegal' much more than background corpus (i.e., liberal and centered media), and it influences on the resulting microframe bias. (C and F), Document (news headline)-level microframe bias spectrum. Red lines indicate headlines from right-wing media, blue lines indicate headlines from left-wing media, and purple lines indicate headlines from center media. Three thick bars with red, blue, and purple colors show the average bias of the conservative, liberal, and center media, respectively. The microframe spectrum effectively shows the microframe bias of news headlines.

Figure 5B shows the top words that contribute the most to the microframe bias on the *illegal-legal* microframe in conservative and liberal media. Conservative media use the word 'illegal' much more than background corpus (i.e., liberal and centered media). Also, they use the word 'amnesty' more frequently, for example, within the context of 'Another Court Strikes Down Obama's Executive Amnesty (Townhall)', and 'patrol' within the context of 'Border Patrol surge as illegal immigrants get more violent (Washington Times)'. Liberal

media mention the word 'illegal' much less than the background and the words 'reform', 'opinion', and 'legal' more.

Figure 5C shows a document (news headline)-level microframe bias spectrum on the *illegal–legal* microframe. The news headline with the highest microframe bias toward 'illegal' is 'U.S. to Stop Deporting Some Illegal Immigrants (Wall Street Journal - News)', and the news headline with the highest microframe bias toward 'legal' is 'Why Trump's Immigration Order is Legal and Constitutional (National Review)'. Compared to a microframe bias-intensity map that shows how individual media use microframes, the microframe bias spectrum makes headline-level microframe biases visible to help to understand which news headlines express what bias.

Considering the second scenario, where we use microframe intensity and bias separation (or microframe intensity and bias compared to a null model) to explore potential microframes in a corpus, we examine the *relaxed –tense* microframe which displays strong intensity separation in news on gun control and gun right issues. Figures 5D–5F show media-, word-, and document-level microframes found by FrameAxis. The average microframe intensity on the *relaxed–tense* microframe is higher in liberal media than conservative media, and the microframe bias of liberal media is toward to 'tense' compared to conservative media. Word-level microframe shift diagrams clearly show that liberal media focus much more on the devastating aspects of gun control, whereas conservative media do not evoke those strong images but focus more on the owner's rights. Figure 5F shows news headlines that are the closest to 'tense.' The key advantage of employing word embedding in FrameAxis is again demonstrated here; out of two headlines in Fig. 5F, none has the word 'tense,' but other words, such as violence or gunfight, deliver the microframe bias toward 'tense.' Although *relaxed–tense* may not be the kind of microframes that are considered in traditional framing analysis, it aptly captures the distinct depictions of the issue in media, opening up doors to further analysis.

The microframe bias-intensity map correctly captures the political leaning of news media as in Fig. 6. Figures 6A and 6C are the microframe bias-intensity maps of news on Democratic party, and Figs. 6B and 6D are those on Republic party. We test the *bad–good* microframe for Figs. 6A and 6B and the *irrational–rational* microframe for Figs. 6C and 6D. The captured bias fits the intuition; Liberal news media show microframe bias toward 'good' and 'rational' when they report the Democratic party, and conservative news media show the same bias when they report the Republican party. Interestingly, their microframe intensity becomes higher when they highlight negative perspectives of those microframes, such as 'bad' and 'irrational.'

As we mentioned earlier, we can also discover relevant microframes given a topic (See *Methods*). As an example, we compute the most relevant microframes given 'abortion' as a topic in Fig. 7. The most relevant microframes to news about 'abortion' indeed capture key dimensions in the abortion debate (*Ginsburg, 1998*). Of course, it is not guaranteed that conservative and liberal media differently use those microframes. The average microframe biases of conservative and liberal media on the four microframes are indeed not statistically different ($p > 0.1$). However, modeling contextual relevance provides a capability to

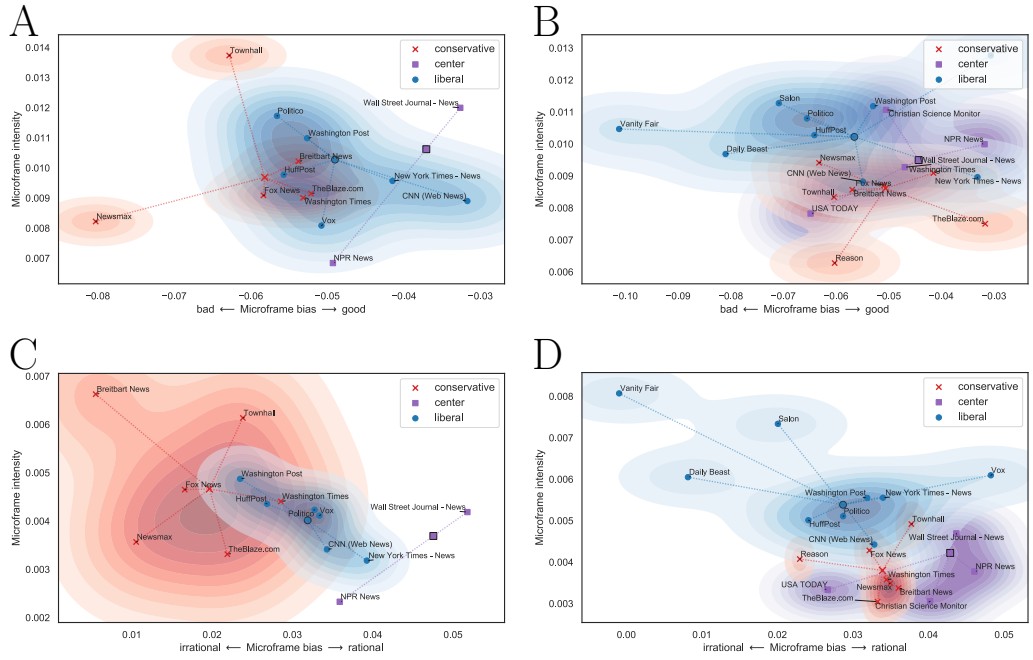

**Figure 6** **Microframe bias-intensity map of (A, C) news on the Democratic party and (B, D) news on the Republican party.** Liberal news media show microframe bias toward 'good' and 'rational' in news on the Democratic party, and conservative news media show microframe bias toward 'good' and 'rational' in news on the Republican party. Microframe intensity of liberal and conservative media is higher when they highlight negative perspectives (i.e., 'bad' and 'irrational').

discover relevant microframes to a given corpus easily even *before* examining the actual data.

## DISCUSSION

In this work, we propose an unsupervised method for characterizing the text by using word embeddings. We demonstrated that FrameAxis can successfully characterize the text through microframe bias and intensity. How biased the text is on a certain microframe (microframe bias) and how actively a certain microframe is used (microframe intensity) provide a nuanced characterization of the text. Particularly, we showed that FrameAxis can support different scenarios: when an important microframe is known (e.g., the *illegal vs. legal* microframe on an immigration issue), when exploration of potential microframes is needed, and when contextually relevant microframes are automatically discovered. The explainability through a document-level microframe spectrum and word-level microframe shift diagram is useful to understand how and why the resulting microframe bias and intensity are captured. They make FrameAxis transparent and help to minimize the risk of spurious correlation that might be embedded in pretrained word embeddings.

We applied FrameAxis to casual texts (i.e., restaurant reviews) and political texts (i.e., political news). In addition to a rich set of predefined microframes, FrameAxis can compute microframe bias and intensity on an arbitrary microframe, so long as it is defined by two

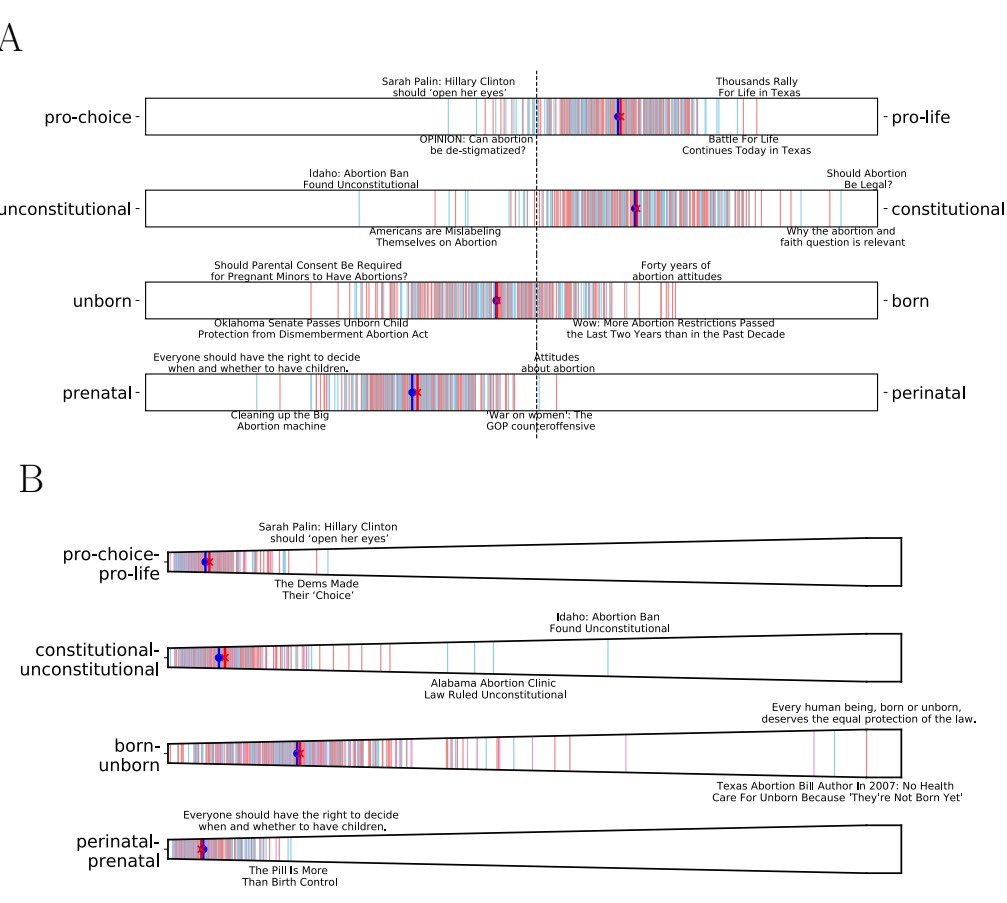

**Figure 7** **Microframe bias and intensity spectra.** (A) Microframe bias spectra and (B) Microframe intensity spectrum of the top 4 most relevant microframes to news about 'Abortion' found by the embedding-based approach. The most relevant microframes indeed capture key dimensions in the abortion debate (*Ginsburg, 1998*).

(antonymous) words. This flexibility provides a great opportunity to study microframes in diverse domains. The existing domain knowledge can be harmoniously combined with FrameAxis by guiding candidate microframes to test and fine-tuning automatically discovered microframes.

Some limitations should be noted. First, word embedding models contain various biases. For example, the word 'immigrant' is closer to 'illegal' than 'legal' (0.463 vs. 0.362) in the GloVe word embedding. Indeed, multiple biases, such as gender or racial bias, in pretrained word embeddings have been documented (*Bolukbasi et al., 2016*). While those biases provide an opportunity to study prejudices and stereotypes in our society over time (*Garg et al., 2018*), it is also possible to capture incorrect microframe bias due to the bias in the word embeddings (or language models). While several approaches are proposed to debias word embeddings (*Zhao et al., 2018*), they have failed to remove those biases completely (*Gonen & Goldberg, 2019*). Nevertheless, since FrameAxis does not depend on specific pretrained word embedding models, FrameAxis can fully benefit from newly

developed word embeddings in the future, which minimize unexpected biases. When using word embeddings with known biases, it may be possible to minimize the effects of such biases through an iterative process as follows: (1) computing microframe bias and intensity; (2) finding top $N$ words that shift the microframe bias and shift; (3) identifying words that reflect stereotypic biases; and (4) replacing those words with an <UNK> token, which is out of vocabulary and thus is not included in the microframe bias and intensity computation, and repeat this process of refinement. The iteration ends when there are no stereotypical words are found in (3). Although some stereotypic biases may exist beyond $N$, depending on $N$, their contribution to microframe shift may be suppressed enough.

Second, an inherent limitation of a dictionary-based approach behind microframe bias and intensity computation exists. Figure 2E reveals the limitation. While 'There was no ambience' conveys a negative connotation, its microframe bias is computed as closer to beautiful than ugly. This error can be potentially addressed by sophisticated end-to-end approaches to model representations of sentences, such as Sentence Transformers (*Reimers & Gurevych, 2019*). While we use a dictionary-based approach for its simplicity and interpretability in this work, FrameAxis can support other methods, including Sentence Transformers, in computing microframe bias and intensity as well. As a proof-of-concept, we use Sentence Transformers to handle the case in Fig. 2E, which is that 'there was no ambiance' has framing bias closer to 'beautiful' than 'ugly'. We compute the representation of three sentences: 'there was no ambience', 'ambience is beautiful', and 'ambience is ugly'. Then, we find that the similarity between 'there was no ambience' and 'ambience is beautiful' (0.3209) is less than that between 'there was no ambience' and 'ambience is ugly' (0.6237). This result indicates that Sentence Transformers correctly understands that the meaning of sentences. As the dictionary-based approach has its own strengths in simplicity and interpretability, future work may seek a way to blend the strengths of different approaches. Even with these limitations, we argue that our approach can greatly help researchers across fields to harness the power of neural embedding methods for text analysis and systematically scale up framing analysis to internet-scale corpora.

We release the source code of FrameAxis, and we will develop it as an easy-to-use library with supporting visualization tools for analyzing microframe bias and intensity for a broader audience. We believe that such efforts would facilitate computational analyses of microframes across disciplines.

### Funding

This research was supported by the Singapore Ministry of Education (MOE) Academic Research Fund (AcRF) Tier 1 grant, Defense Advanced Research Projects Agency (DARPA), contract W911NF17-C-0094, of the United States of America, and the Air Force Office of Scientific Research under award number FA9550-19-1-0391. The funders had no role in study design, data collection and analysis, decision to publish, or preparation of the manuscript.

### Grant Disclosures

The following grant information was disclosed by the authors:

The Singapore Ministry of Education (MOE) Academic Research Fund (AcRF) Tier 1 grant.

Defense Advanced Research Projects Agency (DARPA): contract W911NF17-C-0094.

The Air Force Office of Scientific Research: FA9550-19-1-0391.

### Competing Interests

The authors declare there are no competing interests.

### Author Contributions

- Haewoon Kwak conceived and designed the experiments, performed the experiments, analyzed the data, performed the computation work, prepared figures and/or tables, authored or reviewed drafts of the paper, and approved the final draft.
- Jisun An conceived and designed the experiments, analyzed the data, authored or reviewed drafts of the paper, and approved the final draft.
- Elise Jing performed the computation work, prepared figures and/or tables, and approved the final draft.
- Yong-Yeol Ahn conceived and designed the experiments, authored or reviewed drafts of the paper, and approved the final draft.

### Data Availability

The raw data (political news) and code are available in the Supplemental Files.

### Supplemental Information

Supplemental information for this article can be found online at http://dx.doi.org/10.7717/peerj-cs.644#supplemental-information.

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
