# Peer review of "FrameAxis: characterizing microframe bias and intensity with word embedding"

_PeerJ Computer Science, doi:10.7717/peerj-cs.644_

## Round 0.1 · original submission · Major Revisions

Reviewers raise concerns about how the concept of framing is defined and operationalized in this study. Reviewers (especially 1 & 3) would need to see a stronger motivation to the work, with a (set of) research question(s) properly defined and addressed. More contextualization of the concept of framing, how this is tackled and evaluated, and how it's useful, and to whom, would need to be addressed to improve the paper.

I would encourage authors to submit a major revision if they consider that the above issues can be satisfied.

Reviewer 1 ·

Basic reporting

Please see below

Experimental design

Please see below

Validity of the findings

Please see below

Additional comments

My review here is based on the conceptualization and operationalization of framing. While the effort and intention of designing a new framing method are impressive, I don’t think that the approach suggested by the authors is methodologically sound. Framing is a complicated process and inductively identifying a frame using manual coding is in itself a sophisticated endeavor. Gamson and Modigliani defined frame as a ‘central organizing idea” which “suggests “what the controversy is about, the essence of an issue” (1987). Also, Van Gorp (2010, pp. 91-92) mentions the need to identify reasoning devices like appeals to principles and causal analysis. By relying on microframes, we are ignoring the larger picture which can reduce the meaning of the text by relying on words alone. The authors claim that each microframe is operationalized by an antonym pair, but what about neutral frames? What about generic frames like human interest or economic frames? In the literature, we also have macro frames that are supposed to be the broader overreaching ideas that can be very difficult to automatedly identify.

This approach, unfortunately, will overlook so much meaning. The same argument applies to capturing framing bias. This is again one of the most complicated intellectual tasks for many reasons including the fact that bias is a relative term and operationalizing it using automated methods is almost impossible. What I think the authors are attempting to measure is the sentiments more than the cognitive concepts as they mention the following: “Framing bias in FrameAxis is analogous to positive or negative sentiment in the sentiment analysis, and framing intensity is to polarity, which is the strength of the expressed sentiment”.

In terms of operationalization of the predefined microframes, the authors mentioned that they extracted 1,828 adjective antonym pairs fromWordNetMiller. However, inductive framing analysis is far much complicated than looking at pairs of adjectives. Tankard, for example, referred to 11 framing elements including: 1. Photos, 2. photo captions, 3. Leads, 4. source selection, 5. quotes selection, 6. pull quotes, 7. Logo, 8. statistics and charts, 9. concluding statements and paragraphs, 10. Headlines, 11. Subheads (2001, p. 101). Also, Van Gorp (2010) offers a longer list of framing devices that are similar to the one above, and adjectives are not even amongst them.

To sum up, I think the approach suggested by the authors is interesting because it mixes between sentiment analysis and in a way topic modelling, but it is certainly not related to identifying frames.

Reviewer 2 ·

Basic reporting

The paper satisfies all the requirements.

Experimental design

The method section in the paper is missing some minor details which I point out in the general comments to authors. Again, these are all minor details and can easily be attended to.

Validity of the findings

The work, to the best of my knowledge is novel and the conclusions are well stated.

Additional comments

I would request the authors to explain framing in a bit more detail. To someone coming from Computer science background like me, it was initially a bit difficult to understand what it is. Explaining it with a real world example (may be even from the datasets that you have used) would be helpful. This would certainly improve the readability of the paper.

You mention that while computing framing bias and intensity, you do not consider the topical words. How do you decide which are the topical words? Is it done manually? How many such words did you consider?

The authors also calculate perplexity scores to determine the relevance of a microframe to a given topic. I was not sure how the language model was created. Did you just use a statistical language model and calculate bi-gram probabilities? I would like the authors to elaborate on this a bit.

The captions could be made a bit more informative by adding what one should conclude from them. Since they are all put together at the end of the manuscript, it would be helpful to the reader. The quality of the figures could be improved as well. Caption of figure 4 contains latex symbols.

The references to the figures in discussion section are missing.

Overall, I think it is a nicely written paper and I would be willing to accept it given the above mentioned modifications are made.

Reviewer 3 ·

Basic reporting

This paper proposes a simple approach to detecting prominent aspects of a text corpus, which the authors refer to as "microframes". The method is as follows: i) get a set of antonyms pairs and pretrained word vectors; ii) compute the difference between the word vectors corresponding to each word in a pair, and call this difference a microframe; iii) compute the cosine similarity between each word in the corpus, and report the mean and variance of the cosine similarities for a microframe over all tokens in the corpus.

This is essentially the same idea as one of the earliest papers in bias in word embeddings (Bolukbasi et al., 2016), except that difference vectors are being compared to individual words, rather than other differences, aggregated over tokens.

This is valid way to characterize a corpus, but it is unclear what the ultimate purpose of this sort of approach is. In addition, some of the choices made seem puzzling, and there isn't a very convincing effort to show that this method has advantages over others. Overall, I worry about the usefulness and the robustness of this approach, and exactly what claims the authors are trying to make.

The authors position their approach in comparison to analyses of "framing" in the social sciences. The literature of framing of course is not a monolith. Researchers are often interested in getting deep insight into the arguments that are made in a set of documents, which is why the effort is devoted to developing a codebook and carefully reading the text. It is unclear that the outputs of this system would be of any value to such researchers. Perhaps the authors have a different set of users in mind, but if not social scientists, then whom? (And if social scientists, then more extensive use cases with examples of how this might work in practice would be very helpful.) Stating how users would benefit from this system, and demonstrating that it achieves that goal in comparison to alternative would make this paper more suitable for scholarly publication.

In terms of the reliability of the method, the fact that word vector similarity captures semantic similarity is now well established. The novelty here seems to be partly the use of antonyms from WordNet for coming up with around 1600 potential "microframes". However, I am concerned that many of these will be highly questionable and not informative.

I replicated the authors proposed way of generating such a list, and generated a set of five random pairs, which were:
(quantitative, qualitative)
(incongruent, congruent)
(inessential, essential)
(crosswise, lengthwise)
(agitated, unagitated)

Some of these might be relevant to framing, but it seems like many will not. Moreover, because the authors propose using only two words to establish the direction corresponding to the "microframe", some of these will likely be quite unstable. For example, using the same word vectors as the authors, the most similar words to the qualitative - quantitative direction appear to be: quantitative, Quantitative, QE, QE3, Fed, Bernanke, Treasury.

Presumably texts having to do with "quantiative easing" have given rise to the similarity to (Ben) Bernake and the fed, but this is not related to the qualitative quantitative distinction. More thorough investigation of the potential problems or limitations of this approach would be useful.

A few issues related to the presentation should also be mentioned. There are a few disfluencies, e.g., "the common procedure of the framing research", "corpora" instead of "corpus" when referring to a single corpus, "microaframes" on page 5, "are worth to examine" on page 8, "leveraging word embedding"). There are also improperly compiled references to Figures, such as "Figure ??(E)" on page 10. One extremely annoying problem is that the in-line citations are not formatted properly. They are missing brackets, making them difficult to distinguish from the text.

I am also concerned that Figure A is somewhat misleading. They show cases of high framing intensity as being those when words are similar to the individual poles of the antonym pair. However, because the cosine similarity is being computed with respect to the difference between the pair of vectors, the words that represent the highest intensities should be those that are parallel to the difference between the two. (i.e., starting at the origin, but pointing in the same direction as the difference of vectors). In some cases, the words close to the polls will tend to have high cosine similarity to this vector, but that is more of a property of the word vector space.

As an example, taking a pair the authors highlight -- the (clean, dirty) pair -- the highest intensity words appear to be clean, Clean, ensure, maintain, Excellent, ensure, CLEAN. Clearly some of these are closely aligned with clean (e.g., clean, Clean, CLEAN), but others are not aligned with either clean or dirty individually (ensure, maintain, etc.)

The raw data for one experiment is shared, but not for all. In particular, the full raw results from the human evaluation should also be made available, complete with anonymized worker IDs.

Some terminology is not defined. On page 2 the authors define c_f^w in terms of similarity, but this should be "cosine similarity" to be specific. More importantly, the authors introduce "topic" words on page 3, but do not define what a topic word is, or how they are selected. Beyond that, it is unclear why a topic word would not be relevant to framing, and why they are removed.

Experimental design

The primary experiments in this paper include a Mechanical Turk study, and several visualizations of examples. Although the visualizations are useful for illustrating the kinds of things the authors see this as being used for, they do not do a good job of convincing the read reader that this method is useful.

On the human evaluation, it seems the purpose is to see if the top microframes are more relevant than random ones to the aspect of a restaurant review, but this seems like a very low bar, given that so many of the micorframes appear to be highly niche. (Indeed, I am surprised the results are not stronger here, which makes me wonder about the quality of the annotations). In terms of selecting the significant frames to present, the use of a significance threshold of 0.05, without any correction for multiple comparisons, is surprising, given that there are 1000s of pairs being tested.

The idea of testing the contextual relevance of microframes to a given corpus seems like a poor fit to the rest of the paper, and is not well developed. It is not clear that there is any experimental evaluation of this, and many details are absent. Saying "a language-model based approach" is almost completely uninformative and could refer to a huge range of possibilities. The choice to use particular constructions (A is B) is not well motivated or tested.

Mostly the results consist of displaying lists of microframes that seem to have have face validity (e.g., we assume that hospitable and inhospitable) is relevant to "service", but there is no real evaluation of whether or not this is the "correct" inference for this data or not.

Validity of the findings

As noted above, the usefulness of this method has not been rigorously evaluated. All findings reported seem valid and legitimate, but it is not clear they answer the questions of what is this method useful for, and is it better than the alternatives. Giving more though to the specific scientific claims that the authors want to make would help to determine what experiments are relevant to evaluating those claims. Presenting examples is useful, but not enough on their own.

As also noted above, some raw data have been shared, but not all.

It is unclear what the conclusions of this paper are, in part due to the lack of a clear research question. Although I commend the authors for developing this idea and releasing a system for using it, a research paper based on this idea should have a stronger set of claims being made and well-designed experiments to evaluate those claims.

---

## Round 0.2 · accepted · Accept

Based on the assessment of two reviewers, this paper can now be accepted for publication in its current form.

Reviewer 2 ·

Basic reporting

No comment

Experimental design

No comment

Validity of the findings

No comment

Additional comments

The authors have satisfactorily answered all my questions from the first round of reviews. I have no issues in accepting the paper.

Reviewer 3 ·

Basic reporting

The revised submission of "FrameAxis: Characterizing Microframe Bias and Intensity with Word Embedding" is much improved, and I think it is acceptable for publication.

The key improvements that have been made are as follows:
- reframing of the contribution so as to make it clear that this is a proposed framework for unsupervised exploration of a text corpus, not a way of doing traditional framing analysis
- correction of typos and formatting issues
- improved figures
- clarification of terms like "topic words" and "similarity"
- additional details about certain methodological components, like the use of GPT
- some additional data included with submission

Experimental design

Unlike a traditional computer science paper, the experiments here are much closer to case studies or illustrations, plus sanity checks to show that the method is okay (i.e., the MTurk studies). Everything presented in these case studies seems valid, so by the stated standards of PeerJ, I believe these are acceptable. It is still not obvious that there is a single clear research question, but rather a proposal for a new way to do unsupervised exploration of a textual corpus, with some illustrations of how the more abstract idea of (measuring bias and intensity based on antonym pairs) might be deployed in practice (complete with the many somewhat arbitrary choices that need to be made along the way, like selection of topic words). As such, this is not the kind of paper that contains rigorous experiments demonstrating evidence of new knowledge, nor does it contain quite enough detail to fully replicate everything (for example, precise details on how GPT is being used), but I think they are sufficient to convince people that this approach is something they may want to try for themselves, and enough details and suggestions to help them get started. It is in the nature of this kind of unsupervised exploration that every application is somewhat different, and so it is simply not possible to provide a comprehensive set of parameters and experiments that will work for all settings.

Validity of the findings

The revised paper makes it more clear that the point of the paper is to introduce a new framework for unsupervised exploration of a text corpus, and the case studies show that it has the potential to be useful, in a way that I believe is sound.

Almost all data have been provided. Although the authors did include the MTurk responses with this revision, which is appreciated, they unfortunately did not include the corresponding questions, limiting its usefulness. This answers are still a good thing to share, as it allows others to verify the analysis of these responses, but I would encourage the authors to eventually release the full MTurk data, including questions (either as part of this submission or elsewhere).

The conclusions (discussion) have been updated data, and now more accurately reflect what is illustrated by the paper.

Additional comments

This is a difficult sort of paper to review, because it is aiming at the type of application where individual users will necessarily have to make somewhat arbitrary choices. I do not, for example, think that this paper would meet the standards of a Human-Computer Interaction venue, where there would be a much higher expectation of demonstrating that this system actually helps real users to accomplish some task.

Rather, this paper is more about formalizing an idea that is now well known (using vector similarity based on antonym pairs), and wrapping it in choices that try to turn it into a more general tool that could be used in text analysis. Unfortunately this sort of approach will necessarily involve many somewhat arbitrary choices, such as what counts as a topic word, what antonym pairs should be used, what language model templates are appropriate to use, etc. However, this is likely unavoidable, and I think the current paper does a fair job of illustrating how this method will require these sorts of choices, and of providing reasonable examples of what choices made be made. Again, the correctness of each of these choices is not assessed in this paper, but that is not really the point, since there is not necessarily a "correct" choice in any setting, just a set of choices that will influence the outcome, and which users must be aware of.

I would like to thank the authors for being so receptive to the concerns raised by this and other reviewers, and I too think the paper has been improved considerably via this process.